# Associations of Objectively-Measured Sedentary Time and Patterns with Cognitive Function in Non-Demented Japanese Older Adults: A Cross-Sectional Study

**DOI:** 10.3390/ijerph19041999

**Published:** 2022-02-11

**Authors:** Sanmei Chen, Tao Chen, Takanori Honda, Yu Nofuji, Hiro Kishimoto, Kenji Narazaki

**Affiliations:** 1Department of Global Health Nursing, Graduate School of Biomedical and Health Sciences, Hiroshima University, Hiroshima 734-8553, Japan; chens@hiroshima-u.ac.jp; 2Sports and Health Research Center, Department of Physical Education, Tongji University, Shanghai 200092, China; chentwhy@tongji.edu.cn; 3Department of Epidemiology and Public Health, Graduate School of Medical Sciences, Kyushu University, Fukuoka 812-8582, Japan; honda.takanori.597@m.kyushu-u.ac.jp; 4Research Team for Social Participation and Community Health, Tokyo Metropolitan Institute of Gerontology, Tokyo 173-0015, Japan; nofuji@tmig.or.jp; 5Faculty of Arts and Science, Kyushu University, Fukuoka 819-0395, Japan; kishimoto@artsci.kyushu-u.ac.jp; 6Center for Liberal Arts, Fukuoka Institute of Technology, Fukuoka 811-0295, Japan

**Keywords:** patterns of sedentary time, accelerometer, cognitive function, Japanese, physical activity

## Abstract

This study aimed to investigate the cross-sectional associations of objectively-measured sedentary time and patterns with cognitive function in Japanese older adults. A total of 1681 non-demented community-dwelling older adults (aged 73 ± 6, 62.1% women) were included. Total sedentary time, prolonged sedentary time (accumulated in ≥30 min bouts) and mean sedentary bout length were assessed using a tri-axial accelerometer. Global and domain-specific cognitive functions were measured using the Montreal Cognitive Assessment. The average of total sedentary time and prolonged sedentary time were 462 ± 125 and 186 ± 111 min/day, respectively. Greater prolonged sedentary time, but not total sedentary time, was significantly associated with poorer performance in the orientation domain even after controlling for moderate-to-vigorous physical activity (*p* for trend = 0.002). A significant inverse association was also observed between mean sedentary bout length and the orientation domain (*p* for trend = 0.009). No significant associations were observed for global cognitive function or other cognitive domains. Sedentary time accumulated in prolonged bouts, but not total sedentary time, was inversely associated with orientation ability among older adults. Our results encourage further researches to confirm the role of prolonged sedentary time in changes to cognitive domains over time among older adults.

## 1. Introduction

Cognitive functions such as orientation, memory, and visuospatial abilities, naturally decline with advancing age and often occur in neurodegenerative diseases, particularly dementia [1,2]. Even when a clinical diagnosis of dementia is not reached, declines in cognitive functions are often evident [3] and can cause disability, increased risk of mortality [4], and ultimately substantial increases in healthcare burdens [5]. Increasing evidence from observational studies and randomized clinical trials suggests the benefits of increasing physical activity to promote or maintain cognitive health in later life [6,7]. The 2020 Lancet Commission Report suggests that addressing modifiable risk factors including physical inactivity might prevent or delay up to 40% of dementia cases [8]. Accordingly, it is of critical clinical interest to identify other potentially modifiable risk factors that may delay the progression of cognitive decline.

Sedentary behavior refers to “any waking behavior characterized by an energy expenditure ≤ 1.5 metabolic equivalent units (METs), while in a sitting or reclining posture” [9]. Too much sedentary behavior is distinct from physical inactivity (insufficient amounts of moderate-to-vigorous physical activity, MVPA), as an individual can accumulate large amount of both MVPA (physically active) and sedentary behavior throughout the day [9]. Accumulating evidence in recent years suggested that excess sedentary behavior is highly prevalent in older adults [10,11], and can increase morbidity and mortality risk [12]; however, its association with cognitive function remains inconclusive [13]. This is potentially in part due to the historical reliance on self-reported physical activity questionnaires, which largely focus on leisure-time MVPA, but not sedentary behavior. Thus, the associations of sedentary behavior with cognitive functions remain less understood [14]. As the self-reported measures of sedentary time among older adults are prone to have recall bias, objective measures are increasingly used [15]. To date, several studies have investigated the associations between objectively-measured sedentary time and cognitive functions, but yielded conflicting results [16]. For instance, several studies showed an inverse association that longer sedentary time was associated with poor cognitive functions [17,18], but others found no associations [19,20]. Furthermore, most of those studies merely assessed the associations for the total sedentary time [16]; few studies have addressed the associations for patterns of sedentary time, that is how is sedentary time structured (e.g., sedentary time accumulated in prolonged bouts and bout length) [21].

In the present study, we aimed to investigate the associations of accelerometer-measured total sedentary time and its patterns of accumulation with cognitive functions among non-demented older adults. We hypothesized that sedentary time accumulated in prolonged bouts will be associated with worse cognitive functions.

## 2. Methods

### 2.1. Study Population

The Sasaguri Genkimon Study (SGS), started in 2011, is a community-based prospective cohort study of older adults aged 65 years and over residing in the town of Sasaguri, a suburban town in Fukuoka, Japan [22,23]. The Sasaguri Town has distributions of age, sex, education, and occupation similar to those of the overall Japanese population [24]. As of January 2011, 4979 residents in Sasaguri Town met the SGS inclusion criteria: aged ≥ 65 years or older and not certified as requiring long-term care by Japan’s Long-term Care Insurance System. We contacted all those residents who met the inclusion criteria by sending brochures and questionnaires by mail, after excluding those who had died or moved out of the town by the onset of the study (*n* = 66). Of those 4913 invited people, 2629 subjects consented to participate in the SGS study (participation rate: 53.5%).

For this cross-sectional study, we used data from the baseline survey, which was conducted between May and August, 2011. Some of the subjects were excluded from the present study because of: (i) having a self-reported medical history of dementia or Parkinson’s disease (*n* = 17); (ii) missing or invalid accelerometer data for measurements of sedentary behavior (*n* = 861); or (iii) missing data on measurements of cognitive function (*n* = 37) and covariates (*n* = 33). The final analyses included 1681 participants (640 men and 1048 women), whose mean (standard deviation (SD)) age was 73 ± 6 years (range 65–93). This study was approved by the Institutional Review Board of Fukuoka Institute of Technology. We obtained written informed consent from all participants.

### 2.2. Measurements of Sedentary Behavior

Sedentary time was objectively measured by using a triaxial accelerometer (Active style Pro HJA-350IT; Omron Healthcare, Kyoto, Japan). Participants were instructed to wear the accelerometer on either side of their waist for 7 consecutive days and remove it only for sleeping or water activities, including bathing. We provided a simple, illustrated device-wearing instruction guide and asked participants to write a log diary to increase the compliance to accelerometer protocols. Accelerometer data were considered valid if the participant wore the device for at least 4 valid days (at least 10 h of wear time per day) [25,26].

The intensity of activities was estimated as metabolic equivalent (MET) for every 60 s using built-in algorithms that could identify sedentary activities and classify locomotive and non-locomotive activities [27]. The accuracy of the determination of the MET by the Active style Pro has been validated with the Douglas bag method [27,28]. We used an SAS macro program provided by the National Institute of Cancer to compute non-wear time [29], with modifications for the accelerometer used in our study [30]. Non-wear time was defined as a consecutive period of no activity (i.e., estimated activity intensity < 1.0 METs) for at least 60 min, while allowing for 2 min activities when the intensity rose to 1.0 MET.

Accelerometer-estimated intensity of activities ≤ 1.5 METs were identified as sedentary activities. A sedentary bout was defined as a period in continuous sedentary time where the activity intensity fell into the sedentary range without interruption. Total sedentary time was defined as the total accumulated time spent in all sedentary bouts. We also used two measures to define the patterns of sedentary time: prolonged sedentary time and mean sedentary bout length. The prolonged sedentary time was defined as the total sedentary time accumulated in bouts of ≥30 min [30]. Mean sedentary bout length was calculated as total sedentary time divided by the number of bouts. These three measures were averaged across valid days and expressed as min/day.

### 2.3. Measurements of Cognitive Function

Global cognitive function was assessed by using the Montreal Cognitive Assessment (MoCA), consisting of the following 12 cognitive tasks: a five-item delayed recall task, a clock-drawing task, a cube-copying task, a trail-making task, a phonemic fluency task, a two-item verbal abstraction task, a target-tapping task, a serial subtraction task, a two-item digits-reading task, a three-item naming task, a two-item sentence-repeating task, and a six-item temporal and locational orientation task [31]. The Japanese version of the MoCA has been cross-culturally adapted and validated in the Japanese population [32]. The total score of the MoCA is 30 points, with lower scores indicating poorer global cognitive performance. According to the standard procedure, one point was added to the total score of the MoCA if an individual has 12 years or fewer of formal education ans a MoCA score of less than 30 points. Scores of cognitive domains of the MoCA were also calculated: orientation (6 points); visuospatial abilities (4 points); short-term memory (5 points); executive function (3 points); attention, concentration and working memory (6 points); and language (6 points). We also used the Mini-Mental State Examination (MMSE) to assess probable dementia (MMSE score < 18) [33].

### 2.4. Measurements of Covariates

Data on age and sex were provided by the municipality office. Information on the following covariates was collected by a questionnaire: years of education, living alone (yes or no), current smoking (yes or no), current drinking (yes or no), employed status (yes or no), and self-reported medical history of chronic diseases. Body weight and height were measured using conventional scales. Body mass index (BMI) was calculated by dividing the body weight (kg) by height (m) squared (kg/m^2^). Multimorbidity was defined as the presence of two or more of the 13 following chronic diseases: hypertension, stroke, heart disease, diabetes mellitus, hyperlipidemia, respiratory disease, digestive disease, kidney disease, osteoarthritis or rheumatism, traumatic fracture, cancer, ear disease, and eye disease. Total time spent in MVPA (min/day) was also determined by the tri-axial accelerometer (Active style Pro HJA-350IT, Omron Healthcare, Kyoto, Japan) [24].

### 2.5. Statistical Analysis

The three measures of sedentary time (total sedentary time, prolonged sedentary time, and mean sedentary bout length) were categorized into quartiles. The characteristics of the study participants were summarized according to quartiles of prolonged sedentary time, and presented as means (SD), medians (interquartile range (IQR]), or proportions. We tested trends of the characteristics across quartiles of prolonged sedentary time by using the Jonckheere–Terpstra trend test for continuous variables and the Cochran–Armitage trend test for categorical variables.

We used linear regression models to examine the associations between measures of sedentary time and cognitive functions, since there are no optimal cutoff points for the Japanese version of MoCA [22]. Total sedentary time and prolonged sedentary time were adjusted for accelerometer wear time using the residual method before being categorized into quartiles. Because the total MoCA score and the domain-specific scores were on a different scale, we converted them into Z-scores. We constructed multivariable linear regression models, initially adjusting for age (continuous) and sex (men or women), followed by the variables years of education (continuous), living alone (yes or no), BMI (continuous), multimorbidity (yes or no), employed (yes or no), current smoker (yes or no), and current drinker (yes or no). To examine whether the association was independent of MPVA, we additionally adjusted MVPA (continuous; adjusted by accelerometer wear time using the residual method). We calculated the *p* values for trends by assigning ordinal numbers (0, 1, 2, 3) to each quartile of measures of sedentary time and treating the quartiles as a continuous variable. To account for the multiple testing (3 indicators of sedentary time and 7 outcome variables), we used a conservative approach by applying a Bonferroni corrected threshold for statistical significance (0.05/[3 × 7] = 0.0023). All statistical analyses were performed using SAS software (version 9.4; SAS Institute, Cary, NC, USA).

## 3. Results

The mean (SD) age of participants was 73 (6) years old (range 65–93), and 62.1% were women. On average, participants wore the accelerometer device for 7.1 (1.3) valid days, with an average of 839 (106) min/day. The means (SDs) of total sedentary time and prolonged sedentary time were 462 (125) min/day and 186 (111) min/day, respectively. The mean (SD) sedentary bout length was 8 (3) min. The mean scores (SDs) of cognitive functions were 22.1 (3.7) points for MoCA, 5.8 (0.4) points for orientation, 3.2 (0.8) points for visuospatial ability, 2.1 (1.9) points for short-term memory, 1.9 (0.9) points for executive function, 4.9 (1.1) points for attention, concentration and working memory, and 4.2 (1.1) points for language, respectively.

Table 1 shows the characteristics of participants according to quartiles of prolonged sedentary time (values of prolonged sedentary time were adjusted for accelerometer wear time using the residuals method). Participants who had longer prolonged sedentary time were more likely to be men, older, multimorbid, or smokers, and less likely to be employed. The mean values of BMI, accelerometer wear time, total sedentary time, and mean sedentary bout length increased significantly across ascending quartiles of prolonged sedentary time, while the mean values of MVPA decreased significantly.

Table 2 shows the associations between prolonged sedentary time and performance in global cognitive function and specific cognitive domains. Greater prolonged sedentary time was significantly associated with poorer performance in the orientation domain, even after controlling for MVPA and other potential confounding factors (*p* for trend = 0.002). Prolonged sedentary time was not significantly associated with global cognitive function or other cognitive domains (all *p* for trend > 0.05). No significant associations were observed between total sedentary time and global cognitive function or specific cognitive domains (Appendix A).

Table 3 shows the associations between mean sedentary bout length and performance in global cognitive function and specific cognitive domains. Greater mean sedentary bout length was significantly associated with poorer performance in the orientation domain, even after controlling for MVPA and other potential confounding factors (*p* for trend = 0.009). There were no significant associations of mean sedentary bout length with global cognitive function or other cognitive domains (all *p* for trend > 0.05).

To confirm the robustness of the observed associations for prolonged sedentary time and mean sedentary bout length with the orientation domain, we performed a sensitivity analysis by defining participants who failed to obtain the full six points of orientation tasks as having impairment in the domain of orientation ability (*n* = 181 out of 1681 participants). Accordingly, we used logistic regression analyses to model this binary outcome. The results did not materially change (Appendix A). To diminish the potential bias due to undiagnosed dementia, we repeated the main analyses by excluding probable dementia (MMSE < 18 points) (*n* = 6). We also conducted another sensitivity analysis by controlling for the medical history of each of the 13 pre-existing chronic diseases (hypertension, stroke, heart disease, diabetes mellitus, hyperlipidemia, respiratory dis-ease, digestive disease, kidney disease, osteoarthritis or rheumatism, traumatic fracture, cancer, ear disease, and eye disease) instead of combing them into one covariate as the presence of multimorbidity. The results remained similar in both sensitivity analyses (data not shown).

## 4. Discussion

In this community-based cross-sectional study of non-demented Japanese older adults, prolonged sedentary time in bouts of ≥30 min, but not total sedentary time, showed an inverse association with orientation abilities. This association remained significant even after controlling for MVPA and other potential confounding factors. A significant inverse association was also observed between mean sedentary bout length and the orientation domain. We found no significant associations between any measures of sedentary time and global cognitive function or other cognitive domains. The present study is one of the few to investigate the associations between the accumulation patterns of sedentary time and cognitive function among people who are free of dementia.

The present findings of the accelerometer-measured total sedentary time are reflected within the relevant evidence, which has generally been mixed [16,34]. Some studies reported a statistically significant association between higher objectively measured total sedentary time and poor cognitive function in older adults [17,18], but others reported no such association [19,20,35,36,37]. Moreover, a recent study even reported that accelerometer-assessed total sedentary time was positively associated with cognitive function in healthy middle-aged adults [38]. Differences in inclusion and exclusion criteria, covariates, and cognitive tests may have been attributed to discrepancies in findings across studies. In addition, studies that examined the associations between self-reported different domains of sedentary behavior and cognitive function found TV viewing (which may involve lower cognitive engagement/less cognitively stimulating) was more consistently associated with poorer cognitive function [39,40,41]; while reading or computer use (which may involve higher cognitive engagement) were favorably associated with cognitive function in older adults [39,40]. Thus, it could be speculated that the specific type of activities engaged in while sitting may have opposite effects on cognitive function, which partly explains the absence of association between total sedentary time assessed by accelerometer and cognitive function. Therefore, future studies should assess the specific domains of sedentary behavior, besides total sedentary time, to verify the association between sedentary behavior and cognitive function.

Recently, prolonged sedentary time accumulated in uninterrupted bouts has been linked with detrimental health risks, including poor physical function [42] and a higher risk of cardiovascular disease [43] which are also established bio-makers or risk factors for cognitive function and dementia. Thus, it is plausible that prolonged sedentary time accumulated in continuous bouts would also harm cognitive function. However, in the present study, we observed that greater prolonged sedentary time in bouts of ≥30 min or mean sedentary bout length was significantly associated only with poorer performance in the orientation domain, not with global cognitive function or the other cognitive domains. As mentioned above, cognitive-simulating sedentary activities could be beneficial to cognitive function. Such beneficial effects may counteract the potentially detrimental effect of prolonged sedentary time, which may explain the null results for the global cognitive function in the present study. In addition, the potential mechanism for the detrimental effects of prolonged sedentary time on orientation ability is unclear. Future longitudinal studies with adjunct data on both self-reported and accelerometer-measured sedentary domains are needed to confirm our findings and to better understand the associations between sedentary behaviors and cognitive functions.

One strength of this study is the availability of data on indicators of patterns of sedentary time determined by a tri-axial accelerometer, allowing for the quantification of sedentary time in different specified bouts. Another strength is the use of the Montreal Cognitive Assessment, which is sensitive to subtle cognitive decline as recommended by a recent systematic review [16]. This study also has several limitations. First, we cannot rule out the possibility of undiagnosed dementia among participants included in the final sample. The observed association might have been a result of the potential presence of demented participants. However, the association of prolonged sedentary time with orientation ability remained significant in the sensitivity analysis by excluding probable dementia. Thus, this is unlikely to change our conclusions. Second, the cross-sectional design precludes conclusions about causality. Moreover, although we controlled for many potential confounding factors, residual confounding by unmeasured potential confounding factors, such as dietary factors, remains possible. Longer sedentary time may simply be a marker of a less healthy lifestyle. Finally, we urge caution in generalizing our results to other populations.

## 5. Conclusions

In conclusion, this community-based study of Japanese older adults showed that prolonged sedentary time and mean sedentary bout length, but not total sedentary time, were inversely associated with performance in orientation tasks among older adults, independent of MVPA. Our results encourage further studies to confirm the role of prolonged sedentary time in changes in individual cognitive domains and/or cognitive decline over time among older adults.

## Figures and Tables

**Table 1 ijerph-19-01999-t001:** Characteristics of the study participants by quartiles of prolonged sedentary time (*n* = 1681).

	Prolonged Sedentary Time *				
	Quartile 1 (Low)	Quartile 2	Quartile 3	Quartile 4 (High)	*p* for Trend
Characteristics	(*n* = 420)	(*n* = 420)	(*n* = 421)	(*n* = 420)	
Men, %	16.2	36.2	40.6	58.6	<0.0001
Age, years	71 (5)	73 (6)	73 (6)	75 (6)	<0.0001
Education, years	11.2 (2.1)	11.0 (2.4)	11.1 (2.5)	11.2 (2.7)	0.47
Living alone, %	10.2	14.3	13.3	14.8	0.09
BMI, kg/m^2^	22.5 (2.9)	23.1 (3.2)	23.3 (3.1)	23.8 (3.2)	<0.0001
Multimorbidity, %	40.0	41.4	45.1	50.7	0.0009
Employed, %	23.1	18.6	15.7	13.3	0.0001
Current smoker, %	6.2	7.4	6.2	9.3	0.15
Current drinker, %	34.0	37.6	40.6	43.1	0.005
Accelerometer wear time, min/day	861 (798–918)	828 (757–896)	823 (757–890)	821 (755–900)	<0.0001
Total sedentary time, min/day †	358 (82)	420 (89)	487 (88)	585 (110)	<0.0001
Prolonged sedentary time, min/day †	80 (33)	136 (40)	198 (44)	330 (100)	
Mean sedentary bout length, min †	5.5 (1.0)	6.9 (1.0)	8.5 (1.3)	12.2 (4.0)	<0.0001
MVPA, min/day †	57 (38–85)	44 (28–65)	33 (18–50)	19 (8–38)	<0.0001

Note: Continuous variables are represented as mean (standard deviation) or median (IQR). BMI, body mass index; MVPA, moderate-vigorous physical activity. * The quartile cut-points for prolonged sedentary time were −78, −18, and 58 (for the categorization of quartiles, values of prolonged sedentary time were adjusted for accelerometer wear time by using the residuals method). † Values were not adjusted for accelerometer wear time.

**Table 2 ijerph-19-01999-t002:** Associations between prolonged sedentary time (in bouts of ≥30 min) and performance in global cognitive function and specific cognitive domains (*n* = 1681).

	Unstandardized *β* (95% CI)				*p* Value for Trend
Cognitive Functions	Quartile 1 (Low)	Quartile 2	Quartile 3	Quartile 4 (High)	
Total MoCA score					
Age- and sex-adjusted	reference	0.02 (−0.09 to 0.15)	−0.01 (−0.13 to 0.11)	0.00 (−0.13 to 0.13)	0.87
Multivariable adjusted *	reference	0.03 (−0.08 to 0.15)	−0.01 (−0.12 to 0.11)	−0.02 (−0.14 to 0.11)	0.62
Additionally adjusted for MVPA †	reference	0.04 (−0.08 to 0.16)	0.005 (−0.12 to 0.13)	−0.002 (−0.14 to 0.13)	0.83
Orientation					
Age- and sex-adjusted	reference	−0.03 (−0.14 to 0.09)	−0.05 (−0.17 to 0.06)	−0.18 (−0.30 to −0.06)	0.005
Multivariable adjusted *	reference	−0.05 (−0.16 to 0.07)	−0.08 (−0.20 to 0.04)	−0.22 (−0.34 to −0.09)	<0.001
Additionally adjusted for MVPA †	reference	−0.04 (−0.16 to 0.07)	−0.07 (−0.19 to 0.05)	−0.21 (−0.34 to −0.08)	0.002
Visuospatial abilities					
Age- and sex-adjusted	reference	−0.01 (−0.14 to 0.12)	−0.08 (−0.21 to 0.05)	−0.09 (−0.23 to 0.05)	0.12
Multivariable adjusted *	reference	0.00 (−0.13 to 0.13)	−0.07 (−0.20 to 0.06)	−0.09 (−0.23 to 0.05)	0.12
Additionally adjusted for MVPA †	reference	0.00 (−0.13 to 0.14)	−0.07 (−0.20 to 0.07)	−0.08 (−0.23 to 0.07)	0.18
Short-term memory					
Age- and sex-adjusted	reference	0.11 (−0.01 to 0.24)	−0.03 (−0.16 to 0.10)	0.02 (−0.12 to 0.16)	0.64
Multivariable adjusted *	reference	0.11 (−0.01 to 0.24)	−0.03 (−0.16 to 0.09)	0.00 (−0.13 to 0.14)	0.49
Additionally adjusted for MVPA †	reference	0.12 (−0.01 to 0.25)	−0.02 (−0.15 to 0.12)	0.03 (−0.11 to 0.17)	0.75
Executive function					
Age- and sex-adjusted	reference	0.02 (−0.11 to 0.15)	0.09 (−0.04 to 0.22)	0.02 (−0.12 to 0.16)	0.57
Multivariable adjusted *	reference	0.01 (−0.11 to 0.14)	0.07 (−0.06 to 0.20)	−0.02 (−0.16 to 0.12)	0.99
Additionally adjusted for MVPA †	reference	0.01 (−0.12 to 0.14)	0.06 (−0.07 to 0.20)	−0.04 (−0.19 to 0.11)	0.83
Attention, concentration and working memory					
Age- and sex-adjusted	reference	−0.01 (−0.13 to 0.12)	0.03 (−0.10 to 0.16)	0.07 (−0.07 to 0.21)	0.25
Multivariable adjusted *	reference	0.01 (−0.12 to 0.13)	0.04 (−0.08 to 0.17)	0.08 (−0.06 to 0.21)	0.21
Additionally adjusted for MVPA †	reference	0.01 (−0.12 to 0.14)	0.05 (−0.08 to 0.18)	0.09 (−0.06 to 0.23)	0.19
Language					
Age- and sex-adjusted	reference	−0.06 (−0.19 to 0.06)	0.00 (−0.12 to 0.13)	0.05 (−0.09 to 0.18)	0.34
Multivariable adjusted *	reference	−0.05 (−0.18 to 0.08)	0.01 (−0.12 to 0.14)	0.04 (−0.09 to 0.18)	0.37
Additionally adjusted for MVPA †	reference	−0.04 (−0.17 to 0.08)	0.02 (−0.11 to 0.15)	0.06 (−0.09 to 0.20)	0.29

Note: Prolonged sedentary time (accumulated in bouts of ≥ 30 min) was adjusted for accelerometer wear time. CI, confidence interval; MVPA, moderate-vigorous physical activity. The quartile cut-points for prolonged sedentary time were −78, −18, and 58 (for the categorization of quartiles, values of prolonged sedentary time were adjusted for accelerometer wear time using the residuals method). * Adjusted for age (continuous) and sex (men or women), years of education (continuous), living alone (yes or no), body mass index (continuous), multimorbidity (yes or no), employment (yes or no), current smoker (yes or no), and current drinker (yes or no). † MVPA was adjusted for accelerometer wear time by using the residuals method.

**Table 3 ijerph-19-01999-t003:** Associations between mean sedentary bout length and performance in global cognitive function and specific cognitive domains (*n* = 1681).

	Unstandardized *β* (95% CI)				*p* Value for Trend
Cognitive Functions	Quartile 1 (Low)	Quartile 2	Quartile 3	Quartile 4 (High)	
Total MoCA score					
Age- and sex-adjusted	reference	−0.01 (−0.13 to 0.11)	0.10 (−0.01 to 0.23)	−0.06 (−0.18 to 0.07)	0.83
Multivariable adjusted *	reference	−0.01 (−0.12 to 0.11)	0.09 (−0.02 to 0.21)	−0.05 (−0.17 to 0.07)	0.83
Additionally adjusted for MVPA †	reference	0.00 (−0.12 to 0.11)	0.10 (−0.02 to 0.22)	−0.04 (−0.16 to 0.08)	0.98
Orientation					
Age- and sex-adjusted	reference	0.06 (−0.05 to 0.17)	0.02 (−0.09 to 0.13)	−0.15 (−0.27 to −0.04)	0.01
Multivariable adjusted *	reference	0.04 (−0.07 to 0.16)	0.00 (−0.11 to 0.12)	−0.18 (−0.30 to −0.06)	0.004
Additionally adjusted for MVPA †	reference	0.05 (−0.06 to 0.16)	0.01 (−0.10 to 0.13)	−0.16 (−0.29 to −0.04)	0.009
Visuospatial abilities					
Age- and sex-adjusted	reference	0.03 (−0.10 to 0.16)	0.11 (−0.02 to 0.24)	−0.05 (−0.18 to 0.09)	0.81
Multivariable adjusted *	reference	0.03 (−0.09 to 0.16)	0.10 (−0.03 to 0.23)	−0.03 (−0.16 to 0.11)	0.97
Additionally adjusted for MVPA †	reference	0.04 (−0.09 to 0.17)	0.11 (−0.02 to 0.24)	−0.02 (−0.16 to 0.12)	0.88
Short-term memory					
Age- and sex-adjusted	reference	0.04 (−0.08 to 0.17)	0.04 (−0.08 to 0.17)	−0.04 (−0.17 to 0.09)	0.62
Multivariable adjusted *	reference	0.05 (−0.08 to 0.17)	0.05 (−0.08 to 0.17)	−0.03 (−0.17 to 0.10)	0.64
Additionally adjusted for MVPA †	reference	0.05 (−0.07 to 0.18)	0.05 (−0.07 to 0.18)	−0.02 (−0.16 to 0.12)	0.83
Executive function					
Age- and sex-adjusted	reference	0.01 (−0.12 to 0.14)	0.11 (−0.02 to 0.24)	−0.05 (−0.18 to 0.09)	0.88
Multivariable adjusted *	reference	0.00 (−0.13 to 0.13)	0.08 (−0.04 to 0.21)	−0.07 (−0.21 to 0.06)	0.60
Additionally adjusted for MVPA †	reference	0.00 (−0.13 to 0.12)	0.08 (−0.05 to 0.21)	−0.08 (−0.22 to 0.06)	0.51
Attention, concentration and working memory					
Age- and sex-adjusted	reference	−0.03 (−0.16 to 0.09)	0.10 (−0.02 to 0.23)	0.06 (−0.07 to 0.19)	0.14
Multivariable adjusted *	reference	−0.02 (−0.14 to 0.11)	0.11 (−0.02 to 0.23)	0.08 (−0.05 to 0.22)	0.07
Additionally adjusted for MVPA †	reference	−0.01 (−0.14 to 0.11)	0.11 (−0.01 to 0.24)	0.09 (−0.04 to 0.23)	0.06
Language					
Age- and sex-adjusted	reference	−0.13 (−0.25 to 0.00)	0.01 (−0.12 to 0.14)	−0.05 (−0.18 to 0.08)	0.95
Multivariable adjusted *	reference	−0.13 (−0.25 to 0.00)	0.00 (−0.13 to 0.12)	−0.05 (−0.18 to 0.08)	0.93
Additionally adjusted for MVPA †	reference	−0.13 (−0.25 to 0.00)	0.00 (−0.13 to 0.12)	−0.04 (−0.18 to 0.09)	0.98

Note: CI, confidence interval; MVPA, moderate-vigorous physical activity. The quartile cut-points for mean sedentary bout length were 6.1, 7.6, and 9.5 min. * Adjusted for age (continuous) and sex (men or women), years of education (continuous), living alone (yes or no), body mass index (continuous), multimorbidity (yes or no), employment (yes or no), current smoker (yes or no), and current drinker (yes or no). † MVPA was adjusted for accelerometer wear time using the residuals method.

## Data Availability

Data are not publicly available as the data contain confidential clinical information of the participants, although they are possibly made available from the corresponding author upon reasonable request.

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
