# Peer review of "Associations of Objectively-Measured Sedentary Time and Patterns with Cognitive Function in Non-Demented Japanese Older Adults: A Cross-Sectional Study"

_ijerph, 2022, doi:10.3390/ijerph19041999_

Round 1

Reviewer 1 Report

The text clearly presents a cross-correlation between physical inactivity and cognitive decline. The supplementary tables are interesting and confirm the cross-correlation. However, for this reviewer, correlation is not a "cause and effect" indicator. In this sense, it would be important to evaluate not only the physical activity factor, but also to isolate ALL other factors that can influence the results, such as diet, pre-existing diseases, etc. This reviewer believes that it is essential that such considerations are much clearer in the characterization of the populations analyzed so that the correlations can indicate, more favorably, cause and effect.
This reviewer recommends a review of the results if other factors are not adequately isolated.

Author Response

Comments and Suggestions for Authors

The text clearly presents a cross-correlation between physical inactivity and cognitive decline. The supplementary tables are interesting and confirm the cross-correlation. However, for this reviewer, correlation is not a "cause and effect" indicator. In this sense, it would be important to evaluate not only the physical activity factor, but also to isolate ALL other factors that can influence the results, such as diet, pre-existing diseases, etc. This reviewer believes that it is essential that such considerations are much clearer in the characterization of the populations analyzed so that the correlations can indicate, more favorably, cause and effect.

This reviewer recommends a review of the results if other factors are not adequately isolated.

Response:

Thank you for this suggestion. Indeed, the cross-sectional design precludes causality. Regarding diet or nutritional factors, we could only control BMI in the multivariable-adjusted model, since data on daily diet intake were not available in this study. We, therefore, added the following sentences in the Discussion as one of the limitations of this study:

“Moreover, although we controlled for many potential confounding factors, residual confounding by unmeasured potential confounding factors remains possible, such as dietary factors. Longer sedentary time may simply be a marker of an unhealthier lifestyle. ” (Page 9 Line 298-301)

We have also included multimorbidity as one of the confounding factors. As per suggestion, we conducted a sensitivity analysis by including the medical history of each pre-existing disease instead of multimorbidity. We observed similar results. We, therefore, added the following sentence in the Result:

“We also conducted another sensitivity analysis by controlling for the medical history of each of those 13 pre-existing chronic diseases (hypertension, stroke, heart disease, diabetes mellitus, hyperlipidemia, respiratory dis-ease, digestive disease, kidney disease, osteoarthritis or rheumatism, traumatic fracture, cancer, ear disease, and eye disease) instead of combing them into one covariate as the presence of multimorbidity. The results remained similar.” (Page 8 Line 237-242)

Reviewer 2 Report

1.- In the Discussion you wrote: "The observed association might have been a result of the potential presence of demented participants".

.- I would like, you comment. How did you apply the Exclusion Criteria?, because you wrote at the beginning that the participants were not demented.

2.- You found association between prolonged sedentary time and poorer performance of Orientation domain.

.- It´s interesting, because Orientation in space, time and person is a fundamental cognitive function and the bedrock of neurological and psychiatric mental status examinations.

a.- Brain System for mental orientation en space, time and person.                    NCBI            17 Aug 2015

b.- I advise to read too.

Sina Gerten, Tobias Engeroff, Johannes Fleckenstein, Eszter Füzeki et al.

Deducing the impact of physical Activity, Sedentary Behavior and Physical Performance on Cognitive Function in Healthy Older Adults.

Front. Aging Neurosci., 05 January 2022.

Author Response

Comments and Suggestions for Authors

1.- In the Discussion you wrote: "The observed association might have been a result of the potential presence of demented participants".

.- I would like, you comment. How did you apply the Exclusion Criteria?, because you wrote at the beginning that the participants were not demented.

Response:

Thank you for this comment. As described in the manuscript, we have excluded subjects with needs of long-term care certified by the Long-term Care Insurance system (demented individuals should be certified as requiring long-term care) and those with a self-reported medical history of dementia or Parkinson’s disease. However, there is still a possibility of undiagnosed dementia among participants in the present study. For example, because an older person must contact the municipal government to have the care needs officially certified, some individuals with dementia may have failed to report. Therefore, we have conducted a sensitivity analysis by excluding probable dementia (MMSE < 18 points) (n = 6) and listed this point as one of the limitations of this study.

2.- You found association between prolonged sedentary time and poorer performance of Orientation domain.

.- It´s interesting, because Orientation in space, time and person is a fundamental cognitive function and the bedrock of neurological and psychiatric mental status examinations.

a.- Brain System for mental orientation en space, time and person.                    NCBI            17 Aug 2015

b.- I advise to read too.

Sina Gerten, Tobias Engeroff, Johannes Fleckenstein, Eszter Füzeki et al. Deducing the impact of physical Activity, Sedentary Behavior and Physical Performance on Cognitive Function in Healthy Older Adults. Front. Aging Neurosci., 05 January 2022.

Response:

Thank you for the suggestion. We accordingly cited the later study, as reference No. 37 in the revised manuscript (Page 8 Line 254).

Reviewer 3 Report

Thank you for your manuscript submission: Associations of objectively-measured sedentary time and patterns with cognitive function in non-demented Japanese older adults: a cross-sectional study. The study is of interest and is well written. Only one minor spelling error detected.

Author Response

Comments and Suggestions for Authors

Thank you for your manuscript submission: Associations of objectively-measured sedentary time and patterns with cognitive function in non-demented Japanese older adults: a cross-sectional study. The study is of interest and is well written. Only one minor spelling error detected.

The aim of this paper was to investigate the association between sedentary time and sedentary patterns with cognitive function in Japanese older adults. Total sedentary time, prolonged sedentary time and mean sedentary bout lengths were measured objectively using triaxial accelerometry. Cognitive function were measure using the Montreal Cognitive Assessment. Greater prolonged sedentary time, but not total sedentary time, was significantly associated with poorer performance in the orientation domain even after controlling for moderate to vigorous physical activity. A significant inverse association was also reported between mean sedentary bout length and orientation domain. There is a lack of knowledge and understanding of the role of sedentary time in health and wellbeing and this study has aims to breakdown in detail the patterns of sedentary time which is associated with cognitive function. This study uses objective measures of sedentary time which adds to the robustness of the data collection. This study does contribute to understanding of the role of sedentary time in cognitive function. This study is justified with relevant current literature and conclusions made are appropriate.

Response:

We truly appreciate your positive comments.

Reviewer 4 Report

The subject of study is of great interest given the progressive aging of the population around the world. As the authors mention, studies that provide information on factors that contribute to healthy aging (including good cognitive functioning) have scientific interest.
The methodological proposal is well explained, although the conclusions of this study are limited by the use of a cross-sectional design. The authors mention the interest of conducting longitudinal studies to confirm the results but they do not mention the limitations of their results due to the type of design used.

The following improvements are suggested:
- Better description of  the tables. Specifically in tables 1 and 2, mention what the different quartiles indicate: quartile (low) is low sedentarism time? Or is it the opposite?. Tables should be self-explanatory and make the data easy to read.
- Table 1 - What does age,year mean: 71(5). What does the value "5" mean and successive values in parentheses in that table).

-Present in the discussion section the limitations of this type of design (cross-sectional) to establish conclusions about the association between sedentary lifestyle and cognitive function.

Author Response

Comments and Suggestions for Authors

The subject of study is of great interest given the progressive aging of the population around the world. As the authors mention, studies that provide information on factors that contribute to healthy aging (including good cognitive functioning) have scientific interest.

The methodological proposal is well explained, although the conclusions of this study are limited by the use of a cross-sectional design. The authors mention the interest of conducting longitudinal studies to confirm the results but they do not mention the limitations of their results due to the type of design used.

Response:

We appreciate your positive comments. Indeed, we have listed this point as one of the limitations of this study in Lines 290-292 in our early submission.

The following improvements are suggested:

- Better description of  the tables. Specifically in tables 1 and 2, mention what the different quartiles indicate: quartile (low) is low sedentarism time? Or is it the opposite?. Tables should be self-explanatory and make the data easy to read.

- Table 1 - What does age,year mean: 71(5). What does the value "5" mean and successive values in parentheses in that table).

Response:

Thank you for these comments.  1) We have added cutoffs for each quartile in the footnote in each table in the revision:

“The quartile cut-points for prolonged sedentary time were -78, -18, and 58 (for the categorization of quartiles, values of prolonged sedentary time was adjusted for accelerometer wear time by using the residuals method).” (Page 6 Table 2 Line 207-210)

“The quartile cut-points for mean sedentary bout length were 6.1, 7.6, and 9.5 mins.” (Page 8 Table 3 Line 223-224)

2) We have included “Continuous variables are represented as mean (standard deviation) or median (IQR)” in our early submission. (Page 5 Line 191)

-Present in the discussion section the limitations of this type of design (cross-sectional) to establish conclusions about the association between sedentary lifestyle and cognitive function.

Response:

We have listed this point as one of the limitations of this study (Page 9 Line 297-298).

Other revisions:

To be clear, we added the following notes to Table 1:

“†Values were not adjusted for accelerometer wear time.” (Page 5 Line 194)

Round 2

Reviewer 1 Report

The authors partially answered the questions. The lack of a statistical study between a control group and a target group, leaves the theses pointed out with weak answers. This reviewer understands that there are not enough studies to change the text. Therefore, I partially accept the answers, but I suggest that, in the future, care should be taken so that cross-correlation is not taken as causality.